# Novel, Cost Effective, and Reliable Method for Thermal Conductivity Measurement

**DOI:** 10.3390/s24227269

**Published:** 2024-11-14

**Authors:** Marian Janek, Jozef Kudelcik, Stefan Hardon, Miroslav Gutten

**Affiliations:** 1Department of Physics, Faculty of Electrical Engineering and Information Technology, University of Zilina, Univerzitna 1, 010 26 Zilina, Slovakia; jozef.kudelcik@uniza.sk (J.K.); stefan.hardon@uniza.sk (S.H.); 2Department of Mechatronics and Electronics, Faculty of Electrical Engineering and Information Technology, University of Zilina, Univerzitna 1, 010 26 Zilina, Slovakia; miroslav.gutten@uniza.sk

**Keywords:** thermal conductivity coefficient, low-current source, high-current source

## Abstract

This study describes the development and utilization of a novel setup for measuring the thermal conductivity of polyurethane composites with various nanoparticle contents. Measurements were conducted using both an experimental setup and a professional instrument, the TPS 2500 S, with results demonstrating high agreement with the precision of the measurements. The setup was further validated using a standard reference material with a thermal conductivity of 0.200 W/m/K. Additionally, the reliability of the setup was confirmed by its stability against ambient temperature variations between 20 and 30 degrees Celsius. This research presents a cost-effective method for measuring the thermal conductivity of polyurethane composites. Data processing involves noise reduction and smoothing techniques to ensure reliable results. The setup offers 5% accuracy and proves to be versatile for both research and educational applications.

## 1. Introduction

At the national and European level, as part of the Green Deal agreement, there is an effort to be as ecological as possible and to use low-emission materials from renewable sources, which is one of the goals of our project. Improving the thermal and dielectric properties of new polymer materials compared to the currently used ones, and their modification with nanoparticles, should lead to a longer service life of the individual devices in which they will be used.

Improving the thermal conductivity of polymer-based electrical insulating materials is essential because better heat dissipation enhances the efficiency and longevity of electrical components. It prevents overheating, increases the reliability of devices, and ensures safer operation in applications such as electric machines, transformers, and cables. Effective heat management reduces the likelihood of thermal failure, thus contributing to overall system stability [1,2].

Our research focuses on measuring the thermal properties of polyurethanes with the addition of various types of nanoparticles at different concentrations. We conducted a theoretical study of polyurethane materials, aiming to gain a deeper understanding of their thermal properties [3,4]. These theoretical insights serve as the basis for creating samples of polyurethane with the addition of nanoparticles. To support our theoretical studies, we need experiments with various methods and approaches for measuring thermal properties. This includes the use of commercially available instruments and standard procedures or the development of a custom measurement system due to the unique shape of the samples. Nanoparticles ranging from approximately 10 nm to 100 nm have been shown to improve the thermal properties of polymer materials used as electrical insulators in the electrical industry. These nanoparticles, such as metal oxides, nitrides, or carbides, enhance the thermal conductivity of polymers while maintaining their insulating properties. This leads to better thermal stability and a longer lifespan for electrical components [5,6]. The overall research goal is to assess the impact of adding nanoparticles on the thermal properties of polyurethanes and to obtain important insights for various applications.

Polyurethane samples with nanoparticles are produced by casting from a prepared 3D mold. The technological process determines the sample thickness to be between 1 and 3 mm. To measure the thermal conductivity of materials that are a few millimeters thick, such as our polyurethanes and resins, a few methods are commonly used.

At presence, there are some methods for thermal conductivity coefficient measurements. The Angstrom technique [7] is useful for anisotropic materials or samples with specific geometric constraints. A sinusoidal temperature wave is applied to one end of the sample, and the resulting temperature oscillations are measured at various points. The thermal conductivity is determined from the phase shift and amplitude attenuation of the temperature wave. The thermal diffusivity of polymer films can be studied using the infrared enhanced Angstrom technique [8].

Steady-State Methods include the guarded hot plate and heat flow meter techniques. The guarded hot plate method involves sandwiching the sample between two plates and applying a constant heat flux. The heat flow meter method places the sample between a heat source and a heat sink, and the heat flow is measured across the sample. The thermal conductivity of insulation materials such as polyurethanes can be investigated by steady-state analysis utilizing the Peltier module [9].

Laser Flash Analysis [10] is also effective for materials of varying thickness, including a few millimeters. A short laser pulse heats the front surface of the sample, and an infrared detector measures the temperature rise on the rear surface. The thermal diffusivity obtained can be used to calculate thermal conductivity when the specific heat and density of the material are known.

Transient Plane Source, also known as the hot disk method, is a technique that is suitable for solid materials like resins. The Transient Plane Source sensor, which acts as both a heat source and a temperature sensor, is placed between two pieces of the sample. The temperature increase over time is recorded to determine the thermal conductivity. An investigation of the thermal properties of vacuum insulation panels has been performed in [11]; these results can also be used in the investigation of polyurethanes.

The main drawbacks associated with each of these techniques for measuring the thermal conductivity of polymers include several key factors. The Angstrom Method requires meticulous sample preparation to ensure uniformity and accurate placement of thermocouples, involves complex mathematical models for data interpretation, and demands precise calibration of equipment, including heat sources and sensors. The Steady-State Methods necessitate extended periods to reach thermal equilibrium, leading to longer measurement times, and are sensitive to boundary conditions such as insulation and heat loss, which can affect accuracy. These methods can also involve a complex and rigid setup, making them less adaptable to different sample types and sizes. Laser Flash Analysis requires precise surface preparation to ensure accurate results and involves expensive equipment and complex setups, making the method less accessible for routine measurements. The Transient Plane Source Method requires careful calibration of the sensor for accurate results, can suffer from inaccuracy if the sample acts as a heat sink causing uneven heating, and necessitates sensitive and precise equipment, which can be costly.

Each method’s pros and cons can be summarized as follows:Angstrom Method: Suitable for anisotropic materials but requires precise sample preparation and involves complex mathematical models.Steady-State Methods (Guarded Hot Plate, Heat Flow Meter): Effective for steady thermal conductivity measurements but sensitive to boundary conditions and often requires long measurement times.Laser Flash Analysis: Useful for a variety of thicknesses but demands high-precision equipment and careful surface preparation.Transient Plane Source (Hot Disk Method): Flexible for solid materials, though accuracy may decrease if the sample acts as a heat sink.The physical background is summarized as follows:Fourier’s law describes steady-state heat conduction, applicable in the Guarded Hot Plate and Heat Flow Meter methods.For transient methods, such as the Laser Flash Analysis, heat diffusion is often modeled with one-dimensional transient heat conduction equations, while the Angstrom Method employs phase shift and amplitude attenuation in response to sinusoidal heating.

To create an accurate and cost-effective setup for measuring the thermal conductivity of polymers (polyurethanes) with sample thicknesses of a few millimeters, we developed the following setup. This setup, which achieves an accuracy of 5% without being mathematically complex, features a plane heating source positioned between two samples from the same material of the same thickness and employs two thermistors to monitor the temperature on both sides of the sample. A third sample is placed on top of the two samples to avoid disturbing heat propagation at the interface.

## 2. Setup

### 2.1. Overview of Experimental Setup

A schematic illustrating part of the experimental setup is shown in Figure 1. The heater and thermistors are positioned in designated grooves and connected to an electronic box, which supplies power to the heater element and acquires the analog voltage drop across the thermistors. The electronics box has its own power supply and is connected to a notebook computer to control the data-taking process. The heat pulse is generated by a plane source placed between samples of the investigated polyurethane. First, a thermistor is positioned between the plane source and the sample; second, on the opposite side of the sample, where the second sample covers the second thermistor; then, the third sample is placed on the opposite side of the mentioned setup. The entire assembly, including the samples, heating element, and thermistors, is situated between coolers and secured within a 3D-printed gripper equipped with a digital torque adapter to ensure proper pressure settings. During the generation of the heat pulse and the next measurement time, the thermistors measure temperature at pre-configured time intervals. The thermal conductivity coefficient is calculated from the heat generated, derived from the power flowing through the heating element, the sample dimensions, and the observed temperature difference and corresponding time difference peak spectra. The diagram in Figure 2 visually summarizes the entire experimental setup.

The thermal conductivity coefficient *k* is then obtained by multiplying the power *Q* by the sample thickness *d*, divided by the heating element’s area *S*, temperature ΔT, and time differences Δt in the peaks of the thermistors.
*k* = *Q* * *d*/(*S* * ΔT * Δt).(1)

Measurements commence when the ambient and sample temperatures, as measured by the thermistors, are equal within experimental accuracy. The low-current source for thermistors, the high-current source for the heating element, the data acquisition system, and the thermistor calibration are described in the sections below.

### 2.2. Low-Current Source

Passing a low current through a thermistor is crucial to minimize self-heating effects and ensure accurate temperature measurements. When a current flows through a thermistor, it generates heat, which raises the temperature of the thermistor above the temperature of its environment. This self-heating can introduce significant measurement errors if not properly accounted for. Maintaining a low current helps minimize this self-heating effect and ensures more accurate temperature readings. To ensure the flow of low current through the thermistor, the low-current source described in the next section was designed and implemented.

#### Electrical Scheme

The circuit diagram (see Figure 3) represents a voltage stabilization circuit using a TL431 adjustable shunt regulator and a PNP transistor (BC557B).

The circuit function is as follows. The TL431 is used to regulate the voltage at the base of the BC557B transistor. The capacitor C1 (10 µF) is connected in parallel with the TL431 to filter out any noise and provide stability to the voltage reference. By controlling the base voltage, the transistor operates as a switch to maintain a stable voltage across the load. The high-value resistor (R1) ensures that only a small current flows into the emitter of Q1, which is suitable for our low-current application. The diode (D1) and resistor (R2) combination provide a stable reference voltage to the base of Q1. The output voltage (Uout1) is taken from the collector of Q. The output (Uout) voltage of the thermistor (R3) changes with temperature; the measured voltage is used to calculate the thermistor’s resistance, from which the temperature can be obtained.

### 2.3. High-Current Source for Heating Element

#### 2.3.1. Design

The provided schematic (Figure 4) illustrates a power control circuit using an operational amplifier (OP07), a MOSFET (IRF5Y540CM), and some passive components. The circuit operation is as follows. The op-amp U3 is powered by V3 (+9 V) and V4 (−9 V), while the main power supply for the load is V2 (24 V). The control voltage, V1, is applied to the non-inverting input (+) of the op-amp U3. U3 is configured as a voltage follower (buffer), providing the necessary gate drive voltage to the MOSFET U1. The inverting input (−) of U3 is connected to the source of the MOSFET through the voltage divider formed by R2 and R3. The MOSFET U1 is used as a pass transistor, controlling the current through the R4 (hot disk), with its gate driven by the output of the op-amp U3 through R1. The feedback loop is established from the source of the MOSFET (through R2 and R3) back to the inverting input of the op-amp U3. This feedback ensures that the op-amp adjusts the gate voltage to maintain the desired output voltage across the load. The load current flows from V2 through R4, the drain-source channel of U1, and finally through R3 to ground, with the voltage across R3 providing feedback to the op-amp for stable operation.

The circuit diagram (Figure 5) shows one of the recommended configurations for the TC7660 (see datasheet [12]), which is a charge pump voltage converter IC. The TC7660 is typically used to invert the input voltage when only a positive supply voltage is available.

The description of the circuit is as follows. Capacitor C1 (10 µF) is linked between CAP+ (Pin 1) and CAP− (Pin 3), facilitating the charge pump operation. Ground connections are made between Pin 2 and Pin 6, while Pin 8 receives the positive supply voltage V+. Capacitor C2 (10 µF) is connected between VOUT (Pin 4) and ground, stabilizing the output voltage. The output voltage VOU equals the negative of the input voltage V+.

In our case, a +9 V wall adapter is used as the input voltage. The output of the circuit is positive +9 V and negative −9 V voltage, which are then used to power an operational amplifier (OP07). So, the circuit takes a single positive supply voltage and converts it into a dual positive and negative supply, allowing the use of an operational amplifier (OP07) that requires both positive and negative voltages to properly operate.

#### 2.3.2. Electronic Simulation

The simulation was performed in a freeware program named LTspice [13] and in KiCad [14]. The .dc statement makes a DC sweep analysis, sweeping V1 from 0 to 3.5 V in 0.1 V steps. The .param R = 1 and # .step param R list 1 0.1 0.34 0.5 1 2 statements allow the parameter R to be varied through a list of specified values, used for stepping through different values in a simulation to observe the circuit behavior under various conditions. The circuit acts as a voltage-to-current converter with a 1:1 ratio. The symmetric supply of +9 V and −9 V is crucial. When an asymmetric supply was used (+9 V and Gnd), a nonlinear behavior near zero voltage was observed.

The linear dependence between the input voltage and the voltage on resistor R3 is up to 3.2 V, which corresponds to a current of 3.2 A passing through both resistor R3 and the heating element (R4) is observed (see Figure 6). This circuit is a simple power control circuit where the output voltage (and consequently the current through the load) is controlled by the input control voltage (V1). The op-amp and MOSFET configuration provides a stable and adjustable output suitable for applications requiring precise control of high current loads. The feedback mechanism ensures that the desired output voltage is maintained regardless of variations in the load or power supply.

An Arduino program is used to obtain the power passing through the heating element by measuring the voltage drop across it, considering the resistance of the heating element and the duration of the heat pulse. The pulse duration is set in the Node-RED interface and sent via a serial line to an Arduino Nano. After receiving the signal from Node-RED, the Arduino Nano samples the generated pulse and calculates the power. Then, the measured value is sent through a serial interface to the Node-RED control program.

### 2.4. Data Acquisition

#### 2.4.1. Overview

An Arduino Nano [15] with an ADS1256 24-bit digitizer [16] is used for data acquisition to obtain temperature readings from thermistors by measuring the voltage drop across them. Specifically, the voltage drop is measured across resistor R4, whose precise resistance value is known and measured by an LCR meter (LCR 915) [17]. The current passing through resistor R3 is the same value (precisely selected) as that through the thermistor R4; so by measuring the voltage drop across the thermistor, its resistance can be easily calculated using Ohm’s law.

The measured analog value is converted to a digital value by the ADS1256 24-bit digitizer and sent through an SPI interface to the Arduino Nano, which then transmits the data using a serial line to a notebook (or PC). The timestamp and temperature values calculated from voltage drop are then sent to a notebook which is also used for online control, monitoring, and simple express offline analysis. 

In the offline mode, the acquired data from the thermal conductivity measurement system has been processed using Fast Fourier Transform (FFT) to analyze the frequency components of the temperature signals obtained from the thermistors. FFT is particularly useful in identifying periodic or oscillatory behaviors within the dataset, allowing for a deeper understanding of the thermal response over time. By converting the time domain data into the frequency domain, noise can be filtered out more effectively, and any anomalies in the heat pulse or temperature fluctuations can be identified and corrected. This post-processing step ensures that the final thermal conductivity values are accurate and free from noise interference, enhancing the overall reliability of the measurements.

The difference between offline and online data processing using Fast Fourier Transform (FFT) was found to be much less than 1%, making it negligible. This demonstrates that the real-time, online processing performed during the experiment provided results nearly identical to the more thorough offline analysis. The minimal discrepancy between the two methods indicates that the system’s online data processing is highly reliable and efficient for practical purposes, without compromising the accuracy of the thermal conductivity measurements.

#### 2.4.2. Connection Between ADS1256 Digitizer and Arduino Nano

The ADS1256 is suitable for applications requiring accurate analog measurements, such as load cells, strain gauges, and precision sensors.

Connecting the ADS1256 24-bit analog-to-digital converter (ADC) to the Arduino Nano involves several steps. First, the power pins must be connected, with the VCC pin of the ADS1256 connected to the 3.3 V pin on the Arduino Nano and the DGND pin of the ADS1256 connected to the GND pin on the Arduino Nano. Next, the SPI pins must be connected, with the SCLK pin of the ADS1256 connected to the SCK (pin 13) on the Arduino Nano, the DIN pin of the ADS1256 connected to the MOSI (pin 11) on the Arduino Nano, the DOUT pin of the ADS1256 connected to the MISO (pin 12) on the Arduino Nano, and the CS pin of the ADS1256 connected to any available digital pin on the Arduino Nano, such as pin 10. The DRDY pin of the ADS1256 must also be connected to any available digital pin on the Arduino Nano; in our case, it was pin 2. After these connections were made, the ADS1256 library for Arduino was downloaded and installed from the Arduino Library. The Arduino sketch must then be written, including the ADS1256 library, initializing the ADS1256 object with the CS and DRDY pin numbers, and using the library functions to configure the ADS1256 and read data from the ADC. Finally, the Data Ready (DRDY) interrupt was used to trigger a function that reads data from the ADS1256 when new data are available.

To suppress noise in the ADS1256, several advanced techniques were employed. Firstly, Data Rate Optimization was used. A lower data rate from the range (10–25 Hz) was used to improve the signal-to-noise ratio (SNR). A Programmable Gain Amplifier (PGA) was used to amplify the signal and reduce the noise floor. The PGA can be set to various gain levels, such as 1, 2, 4, 8, 16, 32, or 64; we used an amplification of 16. Also, a Low-Pass Filter was tested to remove high-frequency noise. However, in our case, the effect of the filter was negligible. The Noise Reduction Technique, namely filtering (moving average filter), is implemented in the software on the postprocessing stage for additional noise reduction.

#### 2.4.3. The Arduino Code Description

The code is designed to provide accurate temperature readings from thermistors by utilizing the ADS1256’s high-precision ADC capabilities. It measures the voltage across the thermistors, calculates their resistance, and then determines the temperature based on the Steinhart–Hart equation.

In the Arduino Nano program, the ADS1256 library [18] and standard SPI library are included. Also, constants for the crystal frequency (7.68 MHz) and voltage reference (2.5 V) are defined. The ADS1256 object is created with the specified clock frequency and voltage reference. The function calculates the temperature from the thermistor resistance using the Steinhart–Hart equation. In the setup function, the program initializes serial communication, starts the ADS1256 with a data rate of 10 samples per second and a gain of 16, measures the current through the RED (code name for the first thermistor) and GRAY thermistors (code name for the second thermistor) by averaging multiple readings, and prints the measured currents. In the loop function, the program waits for serial input and upon receiving the “R” character through the serial line from Node-RED [19] controlling the interface that runs on a notebook, measures and prints the timestamp and temperature of the RED thermistor, followed by the GRAY thermistor. For each thermistor, it calculates the resistance from the voltage drop and temperature, and prints these values.

### 2.5. Steinhart–Hart Equation and Calibration

The Steinhart–Hart equation provides excellent curve fitting for NTC (Negative Temperature Coefficient) thermistors over a wide temperature range, typically from −80 °C to 260 °C. In contrast, the simpler Beta equation is useful for narrow temperature ranges and requires only a two-point calibration. However, the more complex Steinhart–Hart equation offers significantly higher accuracy across a wide temperature span, achieved through a three-point calibration and a third-order polynomial model.

Typically, these coefficients are determined experimentally by measuring the thermistor’s resistance at three different known temperatures. By solving a set of three simultaneous equations, the values of A, B, and C can be calculated for a specific thermistor and temperature range or are given by the manufacturer. To obtain the parameters A, B, and C, we perform a fitting procedure for the temperature vs. resistance relation using 16 data points obtained from the datasheet [20], rewriting the Steinhart–Hart equation as follows:T = 1/(A + B ln(R) + C ln (R)^3^),(2)
where T is the absolute temperature in Kelvins, R is the resistance of the thermistor, and A, B, and C are the Steinhart–Hart coefficients.

### 2.6. Programming Graphical User Interface

Figure 7 shows the graphical user interface (GUI) for thermal conductivity measurements. The left and middle sections display two large windows showing the time-dependent temperature readings of thermistors placed near the heating element and on the opposite side of the sample, with the time axis in milliseconds and the temperature axis in degrees Celsius. Next to these windows is the control panel, which includes a settings section for streamlined offline analysis. This panel lists various parameters, such as sample ID, calculated torque for correct sample pressure, sample thickness, hot disk size (square heating element), thermistor diameter, hot disk thickness, resistance values of the hot disk, current, and pulse duration. Based on these parameters, the power and thermal conductivity coefficient can be calculated. The GUI also displays four calculated values: time difference, temperature difference, heat energy from the heating element, and thermal conductivity coefficient. At the bottom, there are three buttons: the first initiates measurement when no current pulse is applied, the second calculates the thermal conductivity coefficient, and the third saves the time-temperature data to a file.

## 3. Results

The thermal conductivity coefficient of pure VUKOL Magna blue and its mixtures with three different concentrations of aluminum nitride (AlN) nanoparticles was evaluated using both our custom-built experimental setup and the professional TPS 2500 S device.

For pure VUKOL Magna blue, our experimental setup measured a thermal conductivity of 0.21 ± 0.01 W/m/K, while the TPS 2500 S measured 0.2096 ± 0.0002 W/m/K. For the sample containing 10% AlN, our setup measured 0.25 ± 0.01 W/m/K, compared to the TPS 2500 S’s measurement of 0.2420 ± 0.0005 W/m/K. In the case of the sample with 20% AlN, our experimental setup recorded a value of 0.28 ± 0.01 W/m/K. This sample was not measured using the TPS 2500 S. Finally, for the sample containing 50% AlN, our setup measured a thermal conductivity of 0.36 ± 0.01 W/m/K, while the TPS 2500 S recorded 0.3725 ± 0.0002 W/m/K.

The 5% accuracy of the setup refers to the maximum deviation of measured thermal conductivity values from those obtained with high-end instruments like the TPS 2500 S, achieved through careful design, component selection, and validation. This accuracy results from optimized components, such as calibrated thermistors and custom low- and high-current sources, which minimize error sources like thermal drift and self-heating. Real-time data processing using Arduino-based acquisition and Node-RED enables precise temperature tracking, with data smoothing and noise reduction further refining measurements. The accuracy was confirmed through comparative testing with the TPS 2500 S, showing consistent alignment across different materials and conditions. For research applications, this 5% accuracy enables detailed studies of materials with low thermal conductivity (starting at 0.200 W/m/K), making it possible to detect subtle differences, even in samples with slight compositional variations. In educational settings, this accuracy allows for effective teaching of thermal conductivity measurement principles with a reliable, accessible setup, providing students with hands-on experience using research-grade measurements. The 5% accuracy thus strikes a balance between affordability and precision, supporting both high-level research and educational use.

For each sample, the thermal conductivity coefficient was measured over 10 times. The measurements were performed at room temperatures ranging from 20 to 30 degrees Celsius. The heating pulse was optimized to remain below 50 degrees Celsius to ensure the thermistor operates within its optimal temperature range, as it provides the highest accuracy up to this temperature. Additional tests were conducted at higher temperatures, reaching up to 100 degrees Celsius, and the calculated thermal conductivity coefficients from these tests were consistent with those obtained within the recommended range, demonstrating the robustness of the setup across a wider temperature span. Moving forward, measurements will be conducted within the thermistor’s recommended temperature range.

The results showed no variation with ambient temperature in this range, demonstrating that the setup is stable and reliable, ensuring consistent and reproducible measurements. The main source of statistical error was attributed to the pressure applied to the sample, which is controlled by a torque adapter. This can be improved by replacing it with a more accurate adapter. The affordability of our experimental setup makes it versatile for various research applications, particularly for studying polyurethanes doped with different nanoparticles, concentrations, and other materials with thermal conductivity starting from 0.200 W/m/K. It also has potential use in student laboratory exercises. The setup’s portability is further enhanced by using the Node-RED control program on a Raspberry Pi, allowing the system to be powered by a battery and easily transportable.

## 4. Discussion

The experimental setup for measuring the thermal conductivity of polyurethane composites, which integrates low-cost hardware and custom-designed electronics is presented. The innovative aspects of our method, in terms of its novel design, cost-effectiveness, and reliability, can be summarized as follows:Our novel setup combines off-the-shelf and custom components in a unique configuration that enables precise thermal conductivity measurements on samples with both standard and non-standard shapes or compositions. This is significant because existing systems often require standard sample geometries and are less adaptable to non-traditional materials. The design integrates a custom gripper, heating element, and thermistor-based sensing within a compact and easily transportable setup. By situating the sample between thermistors on both sides and using a plane heating source, we achieve a more controlled and uniform heating environment, reducing edge effects and improving measurement accuracy. Using Arduino-based data acquisition with high-precision sensors and a flexible control interface (Node-RED) introduces a new level of accessibility and adaptability, allowing for real-time adjustments and control in a way that more rigid commercial systems cannot.The cost-effectiveness of our setup distinguishes it from high-end commercial devices, such as the TPS 2500 S, as it is designed with affordability in mind. By utilizing a simple heating element, an Arduino microcontroller, and standard thermistors, we have significantly reduced material costs without sacrificing accuracy. This affordability opens up thermal conductivity measurement to smaller research facilities, educational institutions, and projects with limited budgets. The custom low and high-current sources are specifically optimized for this setup, avoiding the need for expensive power supplies or specialized equipment typically required in professional-grade systems. By tailoring these sources to meet the exact needs of our measurements, we reduce the costs while maintaining reliability. Furthermore, the entire setup is designed for ease of assembly and reproducibility, meaning it can be built and operated without highly specialized technical skills, lowering the costs associated with training and maintenance.Our setup is reliable, achieving a measurement accuracy within 5% of professional devices—a level of precision that validates its reliability for scientific research and industrial applications. This high accuracy is maintained through careful optimization of components, such as keeping the heating pulse below 50 degrees Celsius to match the optimal range of the thermistors. The design is robust against ambient temperature variations between 20 and 30 degrees Celsius, which we validated through comparative tests with standard reference materials and additional high-temperature tests up to 100 degrees Celsius. This stability across a range of temperatures ensures consistent and reproducible measurements. The Node-RED control system allows for real-time data monitoring and remote adjustments, reducing the chances of errors and enabling precise control over each measurement step. Additionally, the setup includes features for data smoothing and noise reduction, ensuring that results are reliable and free from common measurement interferences.

The setup is particularly well suited for measuring materials with thermal conductivity coefficients as low as 0.200 W/m/K. The portability and ease of use make it ideal for both laboratory research and educational purposes, especially when paired with the Node-RED control interface. Overall, this system presents a versatile and cost-effective solution for measuring thermal conductivity in polymer-based materials, with further potential for improvement in sample pressure control.

## Figures and Tables

**Figure 1 sensors-24-07269-f001:**
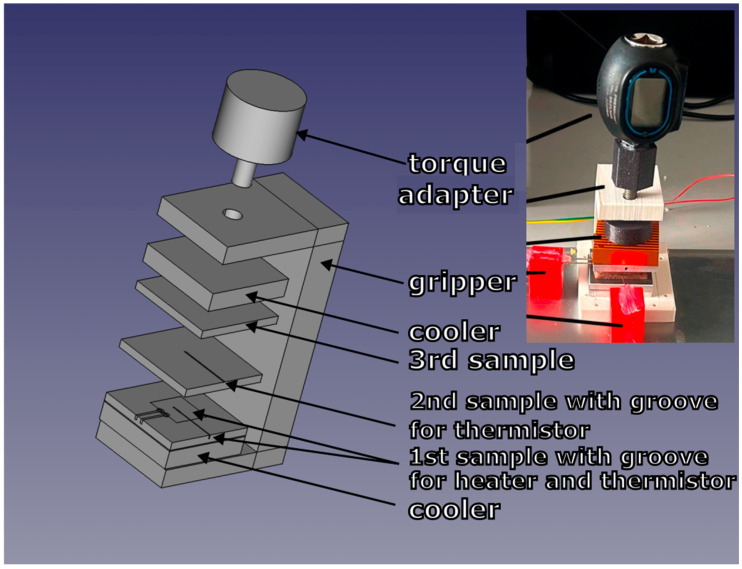
A schematic illustrating part of the experimental setup.

**Figure 2 sensors-24-07269-f002:**
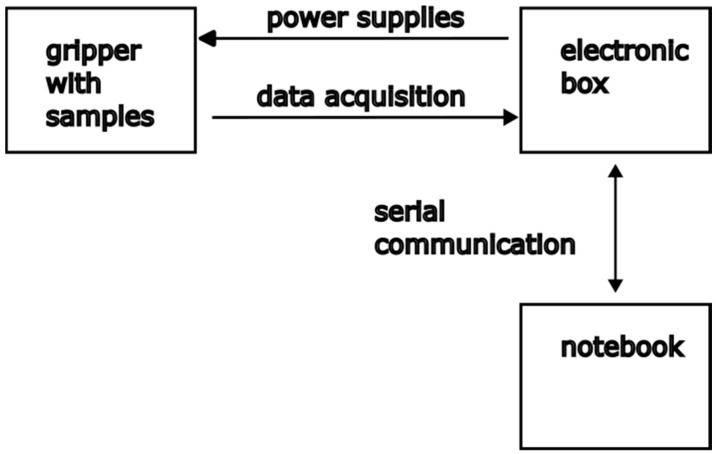
Diagram summarizing the entire experimental setup.

**Figure 3 sensors-24-07269-f003:**
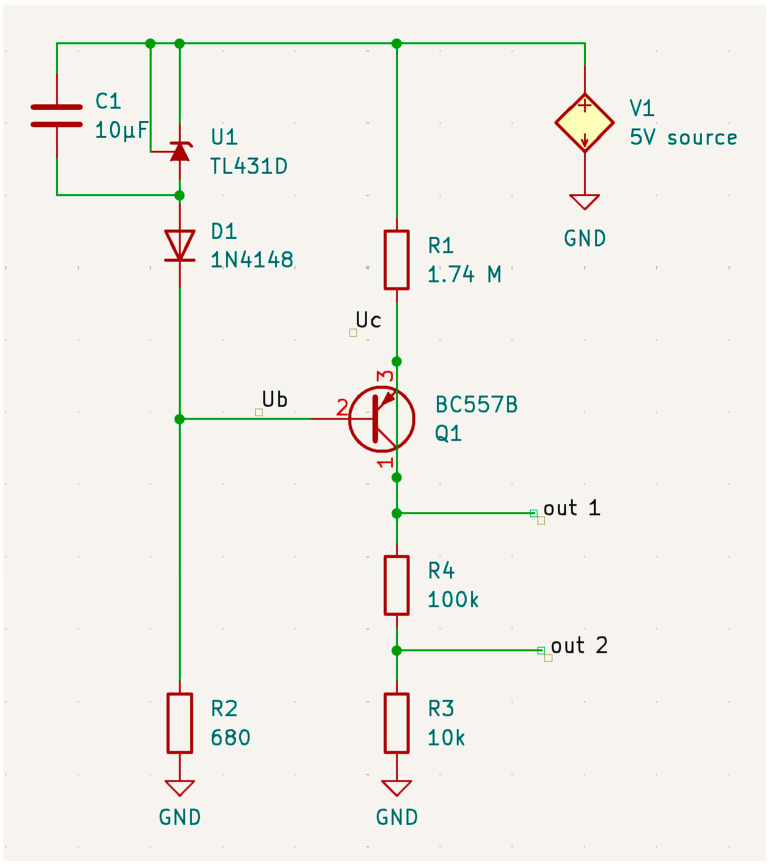
Low-current source scheme for thermistor.

**Figure 4 sensors-24-07269-f004:**
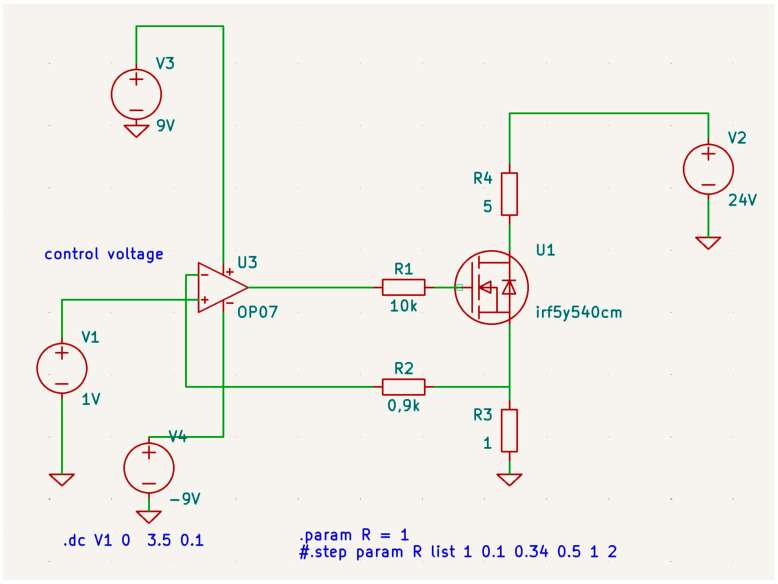
High-current source for R4 heating element.

**Figure 5 sensors-24-07269-f005:**
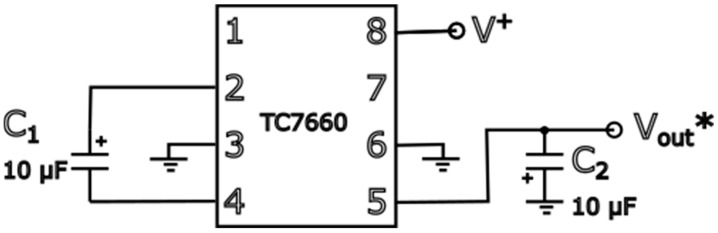
Recommended TC7660 configuration.

**Figure 6 sensors-24-07269-f006:**
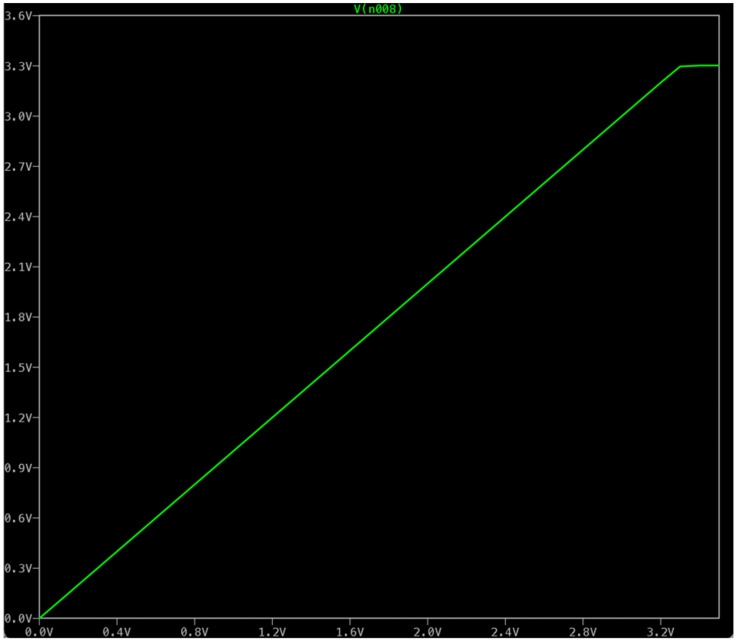
High-current source: linear dependence of input voltage on output voltage (current) up to approximately 3.2 V.

**Figure 7 sensors-24-07269-f007:**
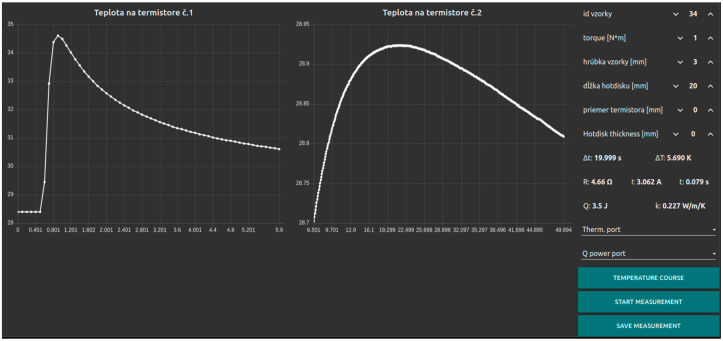
Control programming interface.

## Data Availability

Data are contained within the article.

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
