# Peer review of "Novel, Cost Effective, and Reliable Method for Thermal Conductivity Measurement"

_sensors, 2024, doi:10.3390/s24227269_

Round 1

Reviewer 1 Report

Comments and Suggestions for Authors

The paper presents an innovative and cost-effective setup for measuring the thermal conductivity of polyurethane composites enhanced with nanoparticles. The manuscript addresses a relevant topic in materials science and engineering, especially considering the Green Deal initiative and its emphasis on eco-friendly technologies. While the research is promising, several aspects require further clarification and improvement to strengthen the scientific contribution.

1.    Comparative Overview of Methods:

o   In the introduction, the pros and cons of existing thermal conductivity measurement methods are mentioned. To improve clarity, I recommend adding a table summarizing the comparison. Additionally, please elaborate on the governing equations and physical processes behind these methods.

2.    Experimental Setup Visualization:

o   A schematic or diagram illustrating the experimental setup is essential for better comprehension. A verbal description alone is insufficient.

3.    Physical and Mathematical Foundations:

o   The paper lacks a detailed discussion of the physical principles and mathematical basis for this new method. Please elaborate on this and integrate it into the manuscript.

4.    Applicable Temperature Range:

o   Clarify the temperature range over which the proposed method is effective. This information will enhance the practical relevance of the research.

5.    Circuit Diagram Improvements (Figures 1 & 2):

o   The circuit diagrams appear to be screenshots from simulation software. Consider replotting these diagrams in a more standard format with clear labeling of each component.

6.    Figure 3 Redesign:

o   If Figure 3 is directly copied from a datasheet, please redraw it yourself to avoid copyright issues and align it with the paper’s format.

7.    LTspice Simulation Diagram:

o   In section 2.3.2, a circuit diagram of the LTspice simulation would improve clarity. Please add this to the manuscript.

8.    Experimental Setup Diagram and Photo:

o   A diagram summarizing the entire experimental setup is necessary for better understanding. Additionally, photos of the actual setup is needed.

9.    Figure 4 Cleanup:

o   Figure 4 contains unnecessary information (e.g., browser tabs). Please clean up the image to focus solely on the content relevant to the paper. In addition, Figure. 4 is very difficult to follow.

10.Overemphasis on Circuit Descriptions:

o   The circuits described in the paper are not particularly novel. Please reduce the focus on the circuits and elaborate more on the innovative aspects of your method, e.g. the claimed “Novel, Cost Effective and Reliable” features in the title.

11.Presentation of Experimental Results:

o   The paper currently spends too much space on circuit design. Shift the focus to experimental results. Include figures or plots demonstrating the measured results to provide more insight and validation into the performance of the proposed setup.

Author Response

  1. Comparative Overview of Methods:

o   In the introduction, the pros and cons of existing thermal conductivity measurement methods are mentioned. To improve clarity, I recommend adding a table summarizing the comparison. Additionally, please elaborate on the governing equations and physical processes behind these methods.

Introduction, text added:

To summarize each method’s pros and cons:

  • Angstrom Method: Suitable for anisotropic materials but requires precise sample preparation and involves complex mathematical models.
  • Steady-State Methods (Guarded Hot Plate, Heat Flow Meter): Effective for steady thermal conductivity measurements but sensitive to boundary conditions and often requires long measurement times.
  • Laser Flash Analysis: Useful for a variety of thicknesses but demands high-precision equipment and careful surface preparation.
  • Transient Plane Source (Hot Disk Method): Flexible for solid materials, though accuracy may decrease if the sample acts as a heat sink.

To summarize physical background:

  • Fourier’s law describes steady-state heat conduction, applicable in the Guarded Hot Plate and Heat Flow Meter methods.
  • For transient methods, such as the Laser Flash Analysis, heat diffusion is often modeled with one-dimensional transient heat conduction equations, while the Angstrom Method employs phase shift and amplitude attenuation in response to sinusoidal heating.

  1. Experimental Setup Visualization:

o   A schematic or diagram illustrating the experimental setup is essential for better comprehension. A verbal description alone is insufficient.

The visualization and description of the setup have been added to the Overview of Experimental Setup section.

  1. Physical and Mathematical Foundations:

o   The paper lacks a detailed discussion of the physical principles and mathematical basis for this new method. Please elaborate on this and integrate it into the manuscript.

Text and equation added, see equation 1.

  1. Applicable Temperature Range:

o   Clarify the temperature range over which the proposed method is effective. This information will enhance the practical relevance of the research.

Section results, text added:

The measurements were performed at room temperatures ranging from 20 to 30 degrees Celsius. The heating pulse was optimized to remain below 50 degrees Celsius to ensure the thermistor operates within its optimal temperature range, as it provides the highest accuracy up to this temperature. Additional tests were conducted at higher temperatures, reaching up to 100 degrees Celsius, and the calculated thermal conductivity coefficients from these tests were consistent with those obtained within the recommended range, demonstrating the robustness of the setup across a wider temperature span. Moving forward, measurements will be conducted within the thermistor’s recommended temperature range.

  1. Circuit Diagram Improvements (Figures 1 & 2):

o   The circuit diagrams appear to be screenshots from simulation software. Consider replotting these diagrams in a more standard format with clear labeling of each component.

The visualization and description have been updated.

  1. Figure 3 Redesign:

o   If Figure 3 is directly copied from a datasheet, please redraw it yourself to avoid copyright issues and align it with the paper’s format.

Figure has been updated.

  1. LTspice Simulation Diagram:

o   In section 2.3.2, a circuit diagram of the LTspice simulation would improve clarity. Please add this to the manuscript.

LTspice simulation added.

  1. Experimental Setup Diagram and Photo:

o   A diagram summarizing the entire experimental setup is necessary for better understanding. Additionally, photos of the actual setup is needed.

Photography and diagram added, Figure 1 and 2..

  1. Figure 4 Cleanup:

o   Figure 4 contains unnecessary information (e.g., browser tabs). Please clean up the image to focus solely on the content relevant to the paper. In addition, Figure. 4 is very difficult to follow.

Figure and text has been cleaned up and rewritten for clarity.

10.Overemphasis on Circuit Descriptions:

o   The circuits described in the paper are not particularly novel. Please reduce the focus on the circuits and elaborate more on the innovative aspects of your method, e.g. the claimed “Novel, Cost Effective and Reliable” features in the title.

The detailed description of the circuit has been removed. Novel design, cost-effectiveness, and reliability are discussed in later sections, mainly Discussion.

The innovative aspects of our method, in terms of its novel design, cost-effectiveness, and reliability, can be summarized as follows:

  • Our novel setup combines off-the-shelf and custom components in a unique configuration that enables precise thermal conductivity measurements on samples with both standard and non-standard shapes or compositions. This is significant because existing systems often require standard sample geometries and are less adaptable to non-traditional materials. The design integrates a custom gripper, heating element, and thermistor-based sensing within a compact and easily transportable setup. By situating the sample between thermistors on both sides and using a plane heating source, we achieve a more controlled and uniform heating environment, reducing edge effects and improving measurement accuracy. Using Arduino-based data acquisition with high-precision sensors and a flexible control interface (Node-RED) introduces a new level of accessibility and adaptability, allowing for real-time adjustments and control in a way that more rigid commercial systems cannot.
  • The cost-effectiveness of our setup distinguishes it from high-end commercial devices, such as the TPS 2500 S, as it is designed with affordability in mind. By utilizing a simple heating element, an Arduino microcontroller, and standard thermistors, we have significantly reduced material costs without sacrificing accuracy. This affordability opens up thermal conductivity measurement to smaller research facilities, educational institutions, and projects with limited budgets. The custom low and high current sources are specifically optimized for this setup, avoiding the need for expensive power supplies or specialized equipment typically required in professional-grade systems. By tailoring these sources to meet the exact needs of our measurements, we reduce costs while maintaining reliability. Furthermore, the entire setup is designed for ease of assembly and reproducibility, meaning it can be built and operated without highly specialized technical skills, lowering costs associated with training and maintenance.
  • Our setup is reliable, achieving a measurement accuracy within 5% of professional devices—a level of precision that validates its reliability for scientific research and industrial applications. This high accuracy is maintained through careful optimization of components, such as keeping the heating pulse below 50 degrees Celsius to match the optimal range of the thermistors. The design is robust against ambient temperature variations between 20 and 30 degrees Celsius, which we validated through comparative tests with standard reference materials and additional high-temperature tests up to 100 degrees Celsius. This stability across a range of temperatures ensures consistent and reproducible measurements. The Node-RED control system allows for real-time data monitoring and remote adjustments, reducing the chances of errors and enabling precise control over each measurement step. Additionally, the setup includes features for data smoothing and noise reduction, ensuring that results are reliable and free from common measurement interferences.

11.Presentation of Experimental Results:

o   The paper currently spends too much space on circuit design. Shift the focus to experimental results. Include figures or plots demonstrating the measured results to provide more insight and validation into the performance of the proposed setup.

 The section devoted to electronics is partially cut. We have added the equation, figures, diagram, photograph, and accompanying text to describe the entire method.

Reviewer 2 Report

Comments and Suggestions for Authors

1.    The novelty of this work should be detailed since the title mentioned novelty.

2.    In Page 3 of 12, 2.1 overview of experimental setup. The experimental system of text descriptions is confusing. The author must add corresponding structural diagrams or physical drawings of the experimental system.

3.    The calculation method of thermal conductivity needs to list formulas rather than simple text descriptions. In addition, the author should supplement the calculated results and data of the experiment.

4.  2.2.1 Why “Passing a low current through a thermistor is crucial to minimize self-heating effects and ensure accurate temperature measurements”? What is the relationship between current intensity and temperature measurements? It is necessary to investigate the effect of self-heating effect caused by current on the accuracy of temperature measurement. How to define the intensity of low current? How to determine what level of current intensity is appropriate for temperature measurement?

5.  The author spent many words describing the experimental system and the equipment models involved, and hardly found any relevant experimental data and comparative results. It is difficult to demonstrate the effectiveness and feasibility of this method without the support of experimental data.

6.  Headlines and subheadings are unclear, lack of relevance and logic, making readers feel confused. For example, 2. Materials and Methods. and 2.2. Low current source. and 2.2.1 Purpose.

7.  In the Abstract “The setup offers 5% accuracy and proves versatile for both research and educational applications”. In addition, it is mentioned several times in the article that the method has a 5% accuracy rate. It should be explained further.

Author Response

  1. The novelty of this work should be detailed since the title mentioned novelty.

Added. The novelty and other aspects of the method are discussed in the Discussion section.

  • Our novel setup combines off-the-shelf and custom components in a unique configuration that enables precise thermal conductivity measurements on samples with both standard and non-standard shapes or compositions. This is significant because existing systems often require standard sample geometries and are less adaptable to non-traditional materials. The design integrates a custom gripper, heating element, and thermistor-based sensing within a compact and easily transportable setup. By situating the sample between thermistors on both sides and using a plane heating source, we achieve a more controlled and uniform heating environment, reducing edge effects and improving measurement accuracy. Using Arduino-based data acquisition with high-precision sensors and a flexible control interface (Node-RED) introduces a new level of accessibility and adaptability, allowing for real-time adjustments and control in a way that more rigid commercial systems cannot.

  1. In Page 3 of 12, 2.1 overview of experimental setup. The experimental system of text descriptions is confusing. The author must add corresponding structural diagrams or physical drawings of the experimental system.

A diagram and photo of the experimental setup are added.

  1. The calculation method of thermal conductivity needs to list formulas rather than simple text descriptions. In addition, the author should supplement the calculated results and data of the experiment.

The Overview of Experimental Setup section has been updated, and an equation has been added.

  1. 2.2.1 Why “Passing a low current through a thermistor is crucial to minimize self-heating effects and ensure accurate temperature measurements”? What is the relationship between current intensity and temperature measurements? It is necessary to investigate the effect of self-heating effect caused by current on the accuracy of temperature measurement. How to define the intensity of low current? How to determine what level of current intensity is appropriate for temperature measurement?

Passing a low current through a thermistor is crucial to minimizing self-heating because, as current flows through the thermistor, it generates heat. This self-generated heat raises the temperature of the thermistor above the ambient temperature, causing inaccurate temperature readings.

The heat generated by the current passing through the thermistor is proportional to the square of the current. Higher currents produce more heat, increasing the self-heating effect.

Investigating the self-heating effect is necessary because accurate temperature measurements are critical for precisely locating temperature peaks. Self-heating can distort these measurements, leading to shifts in peak locations and affecting the accuracy of the thermal profile.

Low current for a thermistor is generally defined as the maximum current that avoids significant self-heating, ensuring temperature readings are accurate. We select the lowest current that still allows for precise measurement of the voltage drop across the thermistor. This balance minimizes the self-heating effect while ensuring sufficient sensitivity in the voltage measurement.

  1. The author spent many words describing the experimental system and the equipment models involved, and hardly found any relevant experimental data and comparative results. It is difficult to demonstrate the effectiveness and feasibility of this method without the support of experimental data.

The main goal is to present a setup for thermal conductivity measurement. The results obtained with this setup were compared with the professional apparatus TPS 2500 S, as mentioned in the text. This comparison validates the accuracy and reliability of the developed method.

  1. Headlines and subheadings are unclear, lack of relevance and logic, making readers feel confused. For example, 2. Materials and Methods. and 2.2. Low current source. and 2.2.1 Purpose.

Headlines and subheadings have been updated.

  1. In the Abstract “The setup offers 5% accuracy and proves versatile for both research and educational applications”. In addition, it is mentioned several times in the article that the method has a 5% accuracy rate. It should be explained further.

The 5% accuracy rate of the setup, mentioned in the Abstract and throughout the article, is explained in detail in the Results section.

The 5% accuracy of the setup refers to the maximum deviation of measured thermal conductivity values from those obtained with high-end instruments like the TPS 2500 S, achieved through careful design, component selection, and validation. This accuracy results from optimized components, such as calibrated thermistors and custom low and high current sources, which minimize error sources like thermal drift and self-heating. Real-time data processing using Arduino-based acquisition and Node-RED enables precise temperature tracking, with data smoothing and noise reduction further refining measurements. The accuracy was confirmed through comparative testing with the TPS 2500 S, showing consistent alignment across different materials and conditions. For research applications, this 5% accuracy enables detailed studies of materials with low thermal conductivity (starting at 0.200 W/m/K), making it possible to detect subtle differences, even in samples with slight compositional variations. In educational settings, this accuracy allows for effective teaching of thermal conductivity measurement principles with a reliable, accessible setup, providing students with hands-on experience using research-grade measurements. The 5% accuracy thus strikes a balance between affordability and precision, supporting both high-level research and educational use.

Round 2

Reviewer 1 Report

Comments and Suggestions for Authors

The authors have made corresponding modifications to the issues raised, and the revised paper has reached the level of publication